# Prevalence and genotype distribution of Human Papillomavirus (HPV) among 14,110 women in Anqing urban area: A population-based cross-sectional survey

**Tingting Han[1], Shijie Deng [2]***

**1** Department of Laboratory Medicine, Chaohu Hospital Affiliated to Anhui Medical University, Chaohu, China, **2** Department of Pathology, Anqing First People's Hospital Affiliated to Anhui Medical University, Anqing, China

\* wnmcdsj@qq.com

## Abstract

Human papillomavirus (HPV), particularly persistent infection with high-risk types, is one of the major etiological factors for cervical cancer, posing significan health risks to women. This study aims to analyze the epidemiology of HPV infection among women in the Anqing region, and provide a valuable reference for the prevention and control strategies of cervical cancer in this region. Between 2022 and 2024, a total of 14,110 women attending the First People's Hospital of Anqing were enrolled in this study. All participants underwent both HPV testing and ThinPrep cytology test (TCT). The overall prevalence of HPV infection in the study population was 18.97%. The predominant pattern of HPV infection identified was single infection, followed by double infection. The top five HPV genotypes detected were HPV 52 (2.78%), HPV 81 (1.55%), HPV 58 (1.42%), HPV 16 (1.20%) and HPV 53 (1.16%). The highest HPV positivity rate was observed in women aged < 20 years (79.17%), followed by those aged 50–59 (84.88%), and then those aged > 59 years (42.53%). Among all HPV-positive women, analysis of TCT results revealed that HPV 52, 53, 16, 58 and 81 were frequently detected in ASC-US, HPV 52, 16 and 58 in ASC-H, HPV 52, 58 and 81 in LSIL, and HPV 16, 58 and 18 were the most common genotypes in HSIL. HPV 52, 81, 58, 16 and 53 are the common genotypes among women in Anqing region. The highest prevalence rates of HPV infection were observed in women aged <20 years and those aged 50–59 years. Based on these findings, the 9-valent HPV vaccine is strongly recommended as a primary prevention strategy for this population.

## Introduction

Cervical cancer is a common malignant tumor of the female reproductive tract. In 2020, there were approximately 600,000 new cases and 340,000 related deaths of cervical cancer worldwide, ranking it the fourth most common cancer among women

**Data availability statement:** All relevant data are within the paper and its Supporting Information files.

**Funding:** Fund name: Scientific Research Fund of Anqing Municipal Health Commission in 2024 Fund Number: AQWJ2024011 The funders had no role in study design, data collection and analysis, decision to publish, or preparation of the manuscript.

**Competing interests:** The authors have declared that no competing interests exist. I have read the journal's policy and the authors of this manuscript have the following competing interests: [insert competing interests here].

in terms of both incidence and mortality [1]. However, the incidence of cervical cancer varies significantly across countries, primarily attributable to disparities in economic development, healthcare resource allocation, and public health policies [2–4]. For instance, China, a developing country, cervical cancer caused approximately 109,741 new cases and 59,060 deaths in 2022 [5], imposing substantial clinical and economic burdens. on society. In contrast, the 2023 Surveillance, Epidemiology, and End Results (SEER) Report from the United States indicated that the incidence rate of cervical cancer in 2022 was approximately 6.7 per 100,000 women, reflecting a decline of over 50% in the past two decades due to the widespread implementation of cervical cancer screening programs. As widely recognized, cervical cancer is predominantly caused by high-risk human papillomavirus (Hr-HPV) genotypes infection, which are primarily transmitted through sexual transmission. Several studies have demonstrated that HPV vaccination is highly effective in preventing the development of cervical cancer, particularly against Hr-HPV genotypes such as HPV16 and 18 [6,7]. Although cervical cancer can be effectively prevented through vaccination, HPV vaccines have not yet been widely used in China [8]. Therefore, secondary prevention, namely HPV screening and cervical cytology testing, has become an important strategy for cervical cancer prevention in China.

HPV genotyping has become an essential tool in cervical cancer screening programs worldwide. Unlike traditional cytology alone, HPV genotyping allows for the identification of specific high-risk HPV types, enabling risk stratification and personalized management strategies. Recent advances have also introduced HPV phenotyping, which can detect viral activity and help distinguish between transient infections and those with oncogenic potential. This approach is particularly valuable as a self-sampling method, which gained increased importance during the COVID-19 pandemic when routine healthcare visits were disrupted. The pandemic significantly impacted women's healthcare services globally, leading to delays in cervical cancer screening and follow-up examinations. Studies have shown that the COVID-19 pandemic and the virus infection itself may have affected the quality of medical care and potentially influenced immune responses, which could impact HPV infection persistence and progression [9]. Understanding the epidemiological patterns of HPV during and after this period is crucial for developing effective prevention strategies.

It is notable that the prevalence and genotype distribution of HPV vary across regions, ethnic populations, and sociodemographic strata. However, few information regarding the prevalence of HPV and genotype distribution in the Anqing regions have been reported. Therefore, we conducted retrospective research to analyze the distribution of HPV genotypes and cervical cytology-histology correlation among 14,110 women in southwestern of Anhui province from 2022 to 2024.

## Materials and methods

### Study population and sample collection

The study included 14,110 women recruited from the First People's Hospital of Anqing between January 1, 2022 and December 31, 2024. All participants underwent both human papillomavirus (HPV) testing and ThinPrep liquid-based cytology

(TCT). All the medical records and personal information of the patients were sourced from Anqing First People's Hospital and stored in the hospital's HIS system. Patients were divided into six age groups: <20 years, 20–29 years, 30–39 years, 40–49 years, 50–59 years, and > 59 years. The normal and abnormal cervical lesions diagnosed by cytology were classified into five stages: negative for intraepithelial lesion or malignancy (NILM), atypical squamous cells of undetermined significance (ASC-US), atypical Squamous Cells which cannot exclude high-grade squamous intraepithelial lesion (ASC-H), low-grade squamous intraepithelial neoplasia (LSIL), high-grade squamous intraepithelial neoplasia, and atypical squamous cells – cannot exclude high-grade squamous intraepithelial lesion and cancer(HSIL). Women were enrolled according to the following criteria: (a) had a sexual life history, (b) had no history of immunodeficiency disease, (c) were not pregnant when cervical swabs were collected, (d) had not undergone total uterus or cervix resection, (e) were willing to undergo HPV testing. The study was approved by institutional ethical and research review boards of the participating institutions in the First People's Hospital of Anqing (The Ethical Number: AQYY-YYLY-LWLL-27).

### DNA extraction and HPV genotyping

All cervical swab samples were tested using the HPV GenoArray Test Kit (Triplex International Biosciences, Shenzhen, China) to detect the presence of HPV and determine its genotype. This test can identify 23 HPV genotypes, including 6 low-risk genotypes (HPV 6, 11, 42, 43, 81 and 83) and 17 high-risk genotypes (HPV 16, 18, 31, 33, 35, 39, 45, 51, 52, 53, 56, 58, 59, 66, 68, 73 and 82). The TCT cell preservation solution was produced by Hubei Dellsen Science & Technology Co. Ltd.

### Statistical analysis

Data analysis was performed using SPSS version 22.0 (IBM Corp, Armonk, NY, USA). The prevalence of HPV infection, genotype distribution, and single and multiple HPV infections were analyzed separately. Comparison between single and multiple infections across different cytological grades was performed using Chi-square test. The relative frequencies of HPV genotypes were estimated as percentages. Fisher's chi-square test was used to compare categorical variables, and $P$ values less than 0.05 were considered statistically significant.

## Results

### Overall prevalence of all HPV genotypes

Our study enrolled 14,110 women who underwent HPV testing from January 2022 to December 2024. Among these, 2,677 tested positives for HPV. The overall prevalence of HPV was 18.97%, with positivity rates of 17.79%, 17.65%, and 21.30% in 2022, 2023, and 2024, respectively. The most common HPV genotypes detected were HPV 52 (2.78%), HPV 81 (1.55%), HPV 58 (1.42%), HPV 16 (1.20%) and HPV 53 (1.16%). Among HPV-positive women, single infection was the predominant pattern, followed by double infection. The prevalence of each HPV genotype from 2022 to 2024 is shown in Table 1 and Fig 1.

### Age-specific distribution of HPV infections

According to age, patients were categorized into six age groups: under 20 years, 20–29 years, 30–39 years, 40–49 years, 50–59 years, and over 59 years. The highest HPV positivity rate among total tested women was observed under 20 years old (79.17%, 38/48), followed by those aged 50–59 (84.88%, 3991/4702), and then those aged > 59 years (42.53%, 484/1138). Under 20 years, HPV 11 exhibited the highest prevalence among females, follow by HPV 16. Among women aged 50–59 years, HPV 52 showed the highest positivity rate, follow by HPV 51 and 53. Apart from HPV 83, 18, 39, 45 and 73, the infection rates of other HPV genotypes show statistically significant differences across various age groups (Fig 2 and Table 2).

**Table 1. The overall prevalence of HPV genotypes among 14,110 tested women between 2022 and 2024.**

| | Genotype | 2022 | 2023 | 2024 | total |
|---|---|---|---|---|---|
| Lr-HPV genotype | 6 | 8(0.18) | 11(1.23) | 21(0.42) | 40(0.28) |
| | 11 | 7(0.16) | 10(0.21) | 4(0.08) | 21(0.15) |
| | 42 | 24(0.55) | 22(0.46) | 25(0.51) | 71(0.50) |
| | 43 | 20(0.46) | 24(0.50) | 22(0.45) | 66(0.47) |
| | 81 | 64(1.47) | 63(1.30) | 92(1.86) | 219(1.55) |
| | 83 | 3(0.07) | 1(0.02) | 1(0.02) | 5(0.04) |
| Hr-HPV Genotype | 16 | 59(1.36) | 54(1.12) | 57(1.15) | 170(1.20) |
| | 18 | 23(0.53) | 26(0.54) | 16(0.32) | 65(0.46) |
| | 31 | 9(0.21) | 14(0.29) | 19(0.38) | 42(0.30) |
| | 33 | 27(0.62) | 26(0.54) | 24(0.49) | 77(0.55) |
| | 35 | 3(0.07) | 7(0.14) | 8(0.16) | 18(0.13) |
| | 39 | 16(0.37) | 16(0.33) | 10(0.20) | 42(0.30) |
| | 45 | 3(0.07) | 1(0.02) | 9(0.18) | 13(0.09) |
| | 51 | 31(0.71) | 33(0.68) | 33(0.67) | 97(0.69) |
| | 52 | 110(2.53) | 120(2.48) | 162(3.28) | 392(2.78) |
| | 53 | 41(0.94) | 62(1.28) | 60(1.21) | 163(1.16) |
| | 56 | 20(0.46) | 18(0.37) | 17(0.34) | 55(0.39) |
| | 58 | 52(1.20) | 61(1.26) | 88(1.78) | 201(1.42) |
| | 59 | 16(0.37) | 20(0.41) | 16(0.32) | 52(0.37) |
| | 66 | 10(0.25) | 10(0.21) | 12(0.24) | 32(0.23) |
| | 68 | 26(0.60) | 24(0.50) | 26(0.53) | 76(0.54) |
| | 73 | 1(0.02) | 3(0.06) | 2(0.04) | 6(0.04) |
| | 82 | 4(0.09) | 1(0.02) | 6(0.12) | 11(0.078) |
| Multiple infections | double infection | 141(3.25) | 167(3.45) | 234(4.73) | 542(3.84) |
| | triple infection | 36(0.83) | 35(0.72) | 64(1.29) | 135(0.96) |
| | quadruple infection | 14(0.32) | 16(0.33) | 16(0.32) | 46(0.32) |
| | quintuple infection | 3(0.07) | 8(0.17) | 9(0.18) | 20(0.14) |
| Total | | 771(17.79) | 853(17.65) | 1053(21.30) | 2677(18.97) |

## The relationship between positivity rates of different HPV types and cervical cytology grades

Among all HPV-positive women, distribution analysis of HPV infections across different cytological grades revealed the highest positivity rate in NILM (86.00%), followed by ASC-US (7.62%), ASC-H (2.91%), LSIL (1.83%) and HSIL (1.64%). In women with NILM, the most common HPV genotypes in women were HPV 52, 81, 58 and 53. Among those with ASC-US, the frequently detected HPV genotypes were HPV 52, 53, 16, 58 and 81. For ASC-H, the most common HPV genotypes were HPV 52, 16 and 58. In LSIL, the predominant genotypes among HPV-positive women were HPV 52, 58, and 81. HPV 16, 58 and 18 were the most common HPV genotypes in HSIL (Table 3). We also compared single infections and multiple infections across cytological grades and found that the positivity rate of multiple infections was higher than that of single infections in ASC-H and LSIL (Table 4).

## Discussion

HPV can be categorized into two major groups based on oncogenic risk: low-risk and high-risk genotypes, both of which primarily infect cutaneous and mucosal epithelial tissues. Lr-HPV genotypes predominantly include HPV 6, 11, 42, and 43, whereas Hr-HPV genotypes encompass HPV 16, 18, 31, 51, among others. Substantial evidence has established that

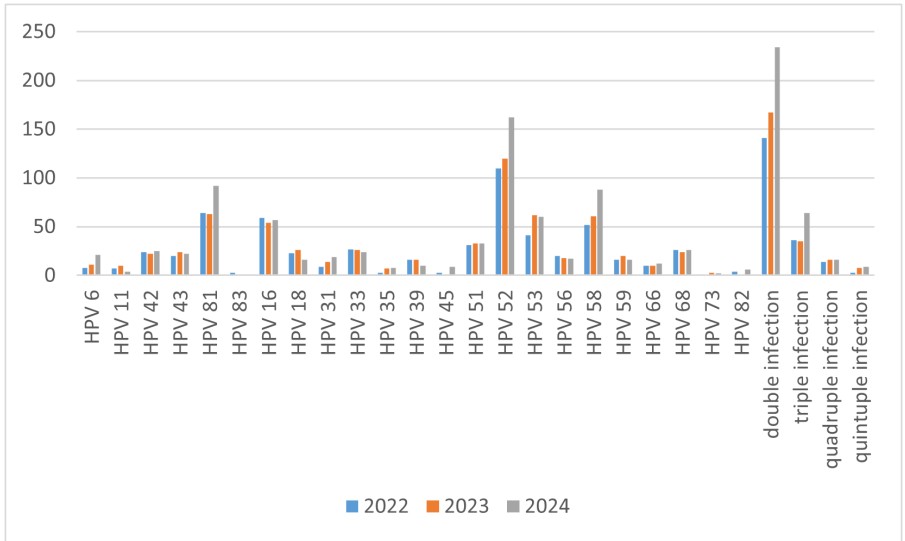

**Fig 1. Overall prevalence of different HPV genotypes.**

cervical carcinogenesis is closely associated with persistent infection with Hr-HPV genotypes, particularly HPV 16 and 18 [10,11]. Notably, the prevalence and genotype distribution of HPV exhibit heterogeneity across populations, regions and vaccination coverage. Hence, determining the prevalence and genotype distribution of HPV in this region holds critical significance for preventing and controlling the development of cervical cancer.

The Anqing area, located in southwestern Anhui Province, is geographically isolated by the Dabie Mountains and characterized by low population mobility. This unique demographic stability enhances the representativeness of our study in reflecting the local HPV epidemiological characteristics within this region, thus providing a more accurate representation of the HPV epidemiological profile within the local population. Between 2022 and 2024, the annual HPV prevalence rates were 17.79%, 17.65% and 21.3%, respectively, with an overall prevalence of 18.97%. The HPV prevalence observed in this study (18.97%) was significantly lower than rates reported in Hangzhou (20.46%) [12], Qujing(24.92%) [13] and Chengdu (25.1%) [14], but higher than those documented in Suzhou (10.2%) [15] and southwestern Yunnan Province (16.95%) [16]. Interestingly, when compared with data from other Asian countries, our findings show similarities with reports from South Korea (17.5%) and Thailand (19.2%), but lower rates than observed in some Southeast Asian regions (25–30%). This regional variation across Asia underscores the importance of local epidemiological studies for tailored prevention strategies [17,18]. Comparison with European data reveals notable differences in HPV genotype distribution. A recent study from Poland utilizing the on clarity test with extended HPV genotyping showed that while HPV 16 and 18 remain predominant in Europe, the prevalence of HPV 52 and 58 is significantly lower compared to our Chinese population [19]. In the Polish cohort of unvaccinated women with suspected squamous intraepithelial lesions, HPV 16 accounted for approximately 35% of high-grade lesions, followed by HPV 31 and HPV 33, whereas in our study, HPV 52 was the most prevalent genotype (23.07% in NILM and 29.90% in ASC-US). This geographic variation in genotype distribution has important implications for vaccine selection and highlights the necessity of the 9-valent vaccine in Asian populations, which provides broader coverage including HPV 52 and 58.

As shown in Fig 1, over the past three years, the most common infection pattern in the Anqing region had been single infection, followed by double infections, with the most common HPV genotypes was HPV 52, 58, 81, and 16. In China, the dominant HPV genotypes vary slightly across different regions. For instance, the main genotypes in Changchun were

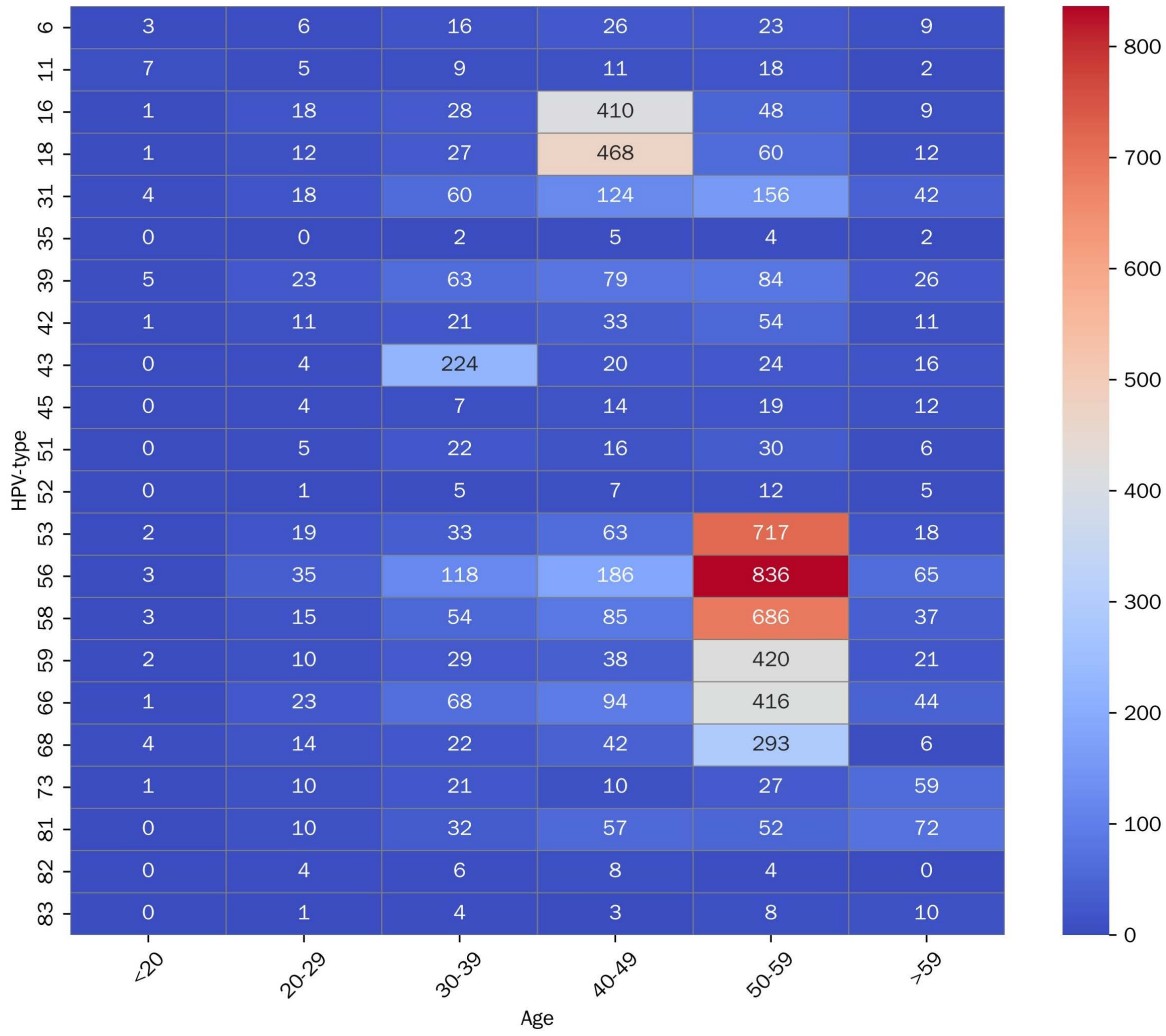

**Fig 2. Heatmap of HPV Infection by Age Group.**

HPV 16, 52, 58, 51 and 53 [20], while in northern Guangdong, they were HPV 52, 16, 58, 53 and 68 [21], and in Luoyang, HPV16, 52, 58, 18 and 51 were the most prevalent Hr-HPV genotype [22]. According to guidelines, the prevalence of HPV infection among female population in Mainland China is estimated to be between 15.5% and 24.3% [23].

The notable increase in HPV prevalence in 2024 (21.30%) compared to 2022 (17.79%) and 2023 (17.65%) warrants discussion. This trend may be attributed to several factors related to the COVID-19 pandemic. First, during the pandemic period (2020−2022), routine cervical cancer screening was significantly disrupted, leading to delayed diagnoses and accumulation of undetected cases. The year 2024 marked a period of healthcare service recovery, with many previously infected women undergoing their first follow-up examination, potentially contributing to the higher detection rate. Second, emerging evidence suggests that the SARS-CoV-2 infection may affect immune function, potentially influencing HPV persistence and reactivation. Third, changes in healthcare-seeking behavior post-pandemic, with increased awareness of preventive health measures, may have led to higher screening participation rates. However, no significant genotype shifts were observed across the study period, suggesting that the increased prevalence primarily reflects screening patterns rather than epidemiological changes in HPV transmission.

**Table 2. Age-specific distribution of HPV infections.**

| Genotype | | <20 (n=48) | 20~29 (n=806) | 30~39 (n=2826) | 40~49 (n=4590) | 50~59 (n=4702) | >59 (n=1138) | P |
|---|---|---|---|---|---|---|---|---|
| Lr-HPV genotype | 6* | 3(6.25) | 6(0.74) | 16(0.57) | 26(0.57) | 23(0.49) | 9(0.79) | 0.002 |
| | 11* | 7(14.58) | 5(0.62) | 9(0.32) | 11(0.24) | 18(0.38) | 2(0.18) | <0.01 |
| | 42* | 1(2.08) | 18(2.23) | 28(0.99) | 410(8.93) | 48(1.02) | 9(0.79) | <0.01 |
| | 43* | 1(2.08) | 12(1.49) | 27(0.96) | 468(10.20) | 60(1.28) | 12(1.05) | <0.01 |
| | 81* | 4(8.33) | 18(2.23) | 60(2.12) | 124(2.70) | 156(3.32) | 42(3.69) | <0.01 |
| | 83 | 0(0) | 0(0) | 2(0.07) | 5(0.11) | 4(0.09) | 2(0.18) | 0.795 |
| Hr-HPV genotype | 16* | 5(10.42) | 23(2.85) | 63(2.23) | 79(1.72) | 84(1.79) | 26(2.28) | 0.001 |
| | 18 | 1(2.08) | 11(1.36) | 21(0.74) | 33(0.72) | 54(1.15) | 11(0.97) | 0.149 |
| | 31* | 0(0) | 4(0.50) | 224(7.93) | 20(0.44) | 24(0.51) | 16(1.41) | <0.01 |
| | 35* | 0(0) | 4(0.50) | 7(0.25) | 14(0.31) | 19(0.40) | 12(1.05) | 0.023 |
| | 39 | 0(0) | 5(0.62) | 22(0.78) | 16(0.35) | 30(0.64) | 6(0.53) | 0.204 |
| | 45 | 0(0) | 1(0.12) | 5(0.18) | 7(0.15) | 12(0.26) | 5(0.44) | 0.41 |
| | 51* | 2(4.17) | 19(2.36) | 33(1.17) | 63(1.37) | 717(15.25) | 18(1.58) | <0.01 |
| | 52* | 3(6.25) | 35(4.34) | 118(4.18) | 186(4.05) | 836(17.78) | 65(5.71) | <0.01 |
| | 53* | 3(6.25) | 15(1.86) | 54(1.91) | 85(1.85) | 686(14.59) | 37(3.25) | <0.01 |
| | 56* | 2(4.17) | 10(1.24) | 29(1.03) | 38(0.83) | 420(8.93) | 21(1.85) | <0.01 |
| | 58* | 1(2.08) | 23(2.85) | 68(2.41) | 94(2.05) | 416(8.85) | 44(3.87) | <0.01 |
| | 59* | 4(8.33) | 14(1.74) | 22(0.78) | 42(0.92) | 293(6.23) | 6(0.53) | <0.01 |
| | 66* | 1(2.08) | 10(1.24) | 21(0.74) | 10(0.22) | 27(0.57) | 59(5.18) | <0.01 |
| | 68* | 0(0) | 10(1.24) | 32(1.13) | 57(1.24) | 52(1.11) | 72(6.33) | <0.01 |
| | 73 | 0(0) | 4(0.05) | 6(0.21) | 8(0.17) | 4(0.09) | 0(0) | 0.111 |
| | 82* | 0(0) | 1(0.12) | 4(0.14) | 3(0.07) | 8(0.17) | 10(0.88) | 0.004 |
| total | | 38(79.17) | 248(30.77) | 871(30.82) | 1799(39.19) | 3991(84.88) | 484(42.53) | |

*P<0.05.

Based on this, it is clinically justifiable to recommend HPV vaccine formulations tailored to local epidemiological patterns. In countries with high HPV vaccination coverage, such as the United States, epidemiological studies have demonstrated a substantial reduction in the prevalence of vaccine-targeted HPV types following large-scale immunization programs. Over a 12-year period post-vaccine introduction (2006–2018), the combined prevalence of HPV 6, 11, 16, and 18 infections declined dramatically from 11.5% (2003–2006) to 1.1%, with the most significant decline first observed among females aged 14–19 years. The HPV infection rate significantly decreased from 12.4% to 4.6% among 25–29 years women and from 9.0% to 4.4% among those aged 30–34 years [24]. In conclusion, population-wide HPV vaccination programs can significantly reduce HPV infection rates at the community level. Based on epidemiological profiles in this region, nine-valent vaccines (HPV 16, 18, 6, 11, 31, 33, 45, 52 and 58) is preferentially recommended over bivalent (HPV 16 and 18) or quadrivalent vaccines (HPV 16, 18, 6 and 11).

In addition to geographical factors, age differences was also a significant influencing factor. Among women of different age groups, the highest positivity rates are observed in those aged 50–59 years and under 20 years, respectively. Our results demonstrated a "two-peak" pattern in Chinese women, with the first peak appearing at youngest age group (15–19 years) and the second peak observed at 50–60 years group [25]. Another peak occurs in women aged 50–59 years, attributed to the viral persistence or reactivation of latent HPV, likely because of the physiological and immunological disorders that can result from hormonal fluctuations during the menopausal transition [26].

**Table 3. The relationship between HPV genotypes and TCT classifications.**

| Genotype | | NILM (n = 2302) | ASC-US (n = 204) | ASC-H (n = 78) | LSIL (n = 49) | HSIL (n = 44) |
|---|---|---|---|---|---|---|
| Lr-HPV genotype | 6 | 72(3.13) | 8(3.92) | 2(2.56) | 1(2.04) | 0(0) |
| | 11 | 38(1.65) | 8(3.92) | 2(2.56) | 2(4.08) | 2(4.55) |
| | 42 | 130(5.65) | 7(3.43) | 2(2.56) | 6(12.24) | 1(2.27) |
| | 43 | 149(6.47) | 8(3.92) | 1(1.28) | 2(4.08) | 0(0) |
| | 81 | 364(15.81) | 22(10.78) | 8(10.26) | 5(10.20) | 0(0) |
| | 83 | 10(0.43) | 2(0.98) | 1(1.28) | 0(0) | 0(0) |
| Hr-HPV Genotype | 16 | 210(9.12) | 29(14.22) | 21(26.92) | 3(6.12) | 17(38.64) |
| | 18 | 108(4.69) | 13(6.37) | 3(3.85) | 3(6.12) | 5(11.36) |
| | 31 | 66(2.87) | 8(3.92) | 0(0) | 2(4.08) | 1(2.27) |
| | 35 | 47(2.04) | 5(2.45) | 1(1.28) | 1(2.04) | 2(4.55) |
| | 39 | 64(2.78) | 8(3.92) | 4(5.13) | 3(6.12) | 0(0) |
| | 45 | 28(1.22) | 0(0) | 0(0) | 0(0) | 2(4.55) |
| | 51 | 167(7.25) | 14(6.86) | 7(8.97) | 9(18.37) | 3(6.82) |
| | 52 | 531(23.07) | 61(29.90) | 24(30.77) | 10(20.41) | 4(9.09) |
| | 53 | 276(11.99) | 31(15.20) | 10(12.82) | 5(10.20) | 1(2.27) |
| | 56 | 126(5.47) | 10(4.90) | 4(5.13) | 5(10.20) | 3(6.82) |
| | 58 | 285(12.38) | 29(14.22) | 16(20.51) | 10(20.41) | 10(22.73) |
| | 59 | 97(4.21) | 6(2.94) | 2(2.56) | 3(6.12) | 2(4.55) |
| | 66 | 51(2.22) | 11(5.39) | 6(7.69) | 5(10.20) | 1(2.27) |
| | 68 | 145(6.30) | 12(5.88) | 3(3.85) | 2(4.08) | 2(4.55) |
| | 73 | 19(0.83) | 1(0.49) | 2(2.56) | 0(0) | 0(0) |
| | 82 | 13(0.56) | 1(0.49) | 2(2.56) | 1(2.04) | 2(4.55) |

**Table 4. Comparison of single infections and multiple infections across cytological grades.**

| Types of Infection | total | NILM | ASC-US | ASC-H | LSIL | HSIL |
|---|---|---|---|---|---|---|
| single infections | 1936 | 1708(88.22%) | 124(7.26%) | 46(2.38%) | 26(1.34%) | 32(1.65%) |
| Multiple infections | 741 | 607(81.92%) | 33(4.45%) | 64(8.64%) | 25(3.37%) | 12(1.62%) |
| $\chi^2$ | | 18.228 | 3.697 | 53.313 | 11.827 | 0.004 |
| P | | <0.001 | 0.055 | <0.001 | 0.001 | 0.951 |

Additionally, this study also analyzed the relationship between TCT results and HPV infection. Among patients with abnormal TCT results requiring intervention, the Hr-HPV subtypes with higher positivity rates in the ASC-US and ASC-H groups were HPV 52, 16, 58, and 53, while the Lr-HPV subtype was HPV 81. In the LSIL group, the most prevalent subtypes were HPV 52, 58, and 51. HPV 16 and HPV 58 were the most common subtypes in the HSIL group. Globally, HPV 16 and HPV 18 account for approximately 70% of all cervical cancers [15], whereas HPV 52 and HPV 58 are more prevalent among women with abnormal TCT results in China [27]—a pattern consistent with our findings.

It is notable that HPV is not only associated with cervical cancer but also serves as a significant causative factor in the development of oropharyngeal cancer [28]. Recent global trends indicate an increasing prevalence of oral HPV infection, particularly HPV 16, which is now recognized as a major driver of oropharyngeal squamous cell carcinoma in many developed countries. In Asia, including China, the epidemiology of oral HPV remains understudied compared to cervical HPV. Given the rising incidence of HPV-associated oropharyngeal cancers globally and the potential for cross-site transmission, future investigations should expand beyond cervical screening to include oral HPV surveillance in the Chinese population.

Understanding oral HPV prevalence and its relationship to cervical HPV infection patterns would provide valuable insights for comprehensive HPV prevention strategies and potentially inform vaccine recommendations for both sexes [17,18,29].

Moreover, there is an increasing consensus and evidence for the association of various viruses with Glioblastoma multiforme (GBM), and HPV is believed to potentially contribute to the development of certain brain tumors, playing an important role in lung cancer and cutaneous squamous cell carcinoma [30]. Beyond Hr-HPV genotypes, Lr-HPV genotypes including HPV 81, 11 and 42 should receive attention due to their association with benign proliferative lesions [31]. Therefore, future vaccine development should prioritize formulations targeting specific Lr-HPV genotypes to expand protection against associated benign pathologies, as recommended in the WHO technical guidance on HPV vaccine standardization (2023) [32].

## Limitations

Several limitations of this study should be acknowledged. First, the retrospective design limits our ability to assess temporal relationships and causality. Second, vaccination history was incomplete for a portion of the cohort, which may affect the interpretation of prevalence data. Third, the study was conducted at a single hospital, which may limit generalizability to rural areas or other regions of China. Fourth, we did not perform viral load quantification or assess HPV persistence, which are important factors in cervical cancer risk stratification. Finally, the impact of COVID-19 on immune function and HPV natural history remains speculative and requires prospective studies with immunological assessments.

## Conclusion

In summary, this study examined the epidemiological characteristics of HPV infection, including prevalence and genotype distribution, among women in the southwestern region of Anhui Province, China. Based on our findings, females at an elevated risk of HPV exposure are strongly recommended to receive prioritized administration of the nine-valent HPV vaccine.

## Supporting information

**S1 Table. The overall prevalence of HPV genotypes among 14,110 tested women between 2022 and 2024.** (PDF)

**S2 Table. Age-specific distribution of HPV infections.** (PDF)

**S3 Table. The relationship between HPV genotypes and TCT classifications.** (PDF)

**S4 Table. Comparison of single infections and multiple infections across cytological grades.** (PDF)

## Acknowledgments

We sincerely thank the staff of the Department of Pathology at the First People's Hospital of Anqing for providing epidemiological investigation support. We also acknowledge the impact of the COVID-19 pandemic on healthcare services and thank all healthcare workers who maintained cervical cancer screening programs during this challenging period.

## Author contributions

**Data curation:** Tingting Han.

**Formal analysis:** Tingting Han.

**Funding acquisition:** Shijie Deng.

**Methodology:** Shijie Deng.

**Supervision:** Shijie Deng.

**Writing – original draft:** Tingting Han.

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
