## [Decision Letter · Decision Letter 0]

7 Oct 2025

Dear Dr. Deng,

Thank you for submitting your manuscript to PLOS ONE. After careful consideration, we feel that it has merit but does not fully meet PLOS ONE’s publication criteria as it currently stands. Therefore, we invite you to submit a revised version of the manuscript that addresses the points raised during the review process.

Please submit your revised manuscript by Nov 21 2025 11:59PM. If you will need more time than this to complete your revisions, please reply to this message or contact the journal office at plosone@plos.org . A rebuttal letter that responds to each point raised by the academic editor and reviewer(s). You should upload this letter as a separate file labeled 'Response to Reviewers'.A marked-up copy of your manuscript that highlights changes made to the original version. You should upload this as a separate file labeled 'Revised Manuscript with Track Changes'.An unmarked version of your revised paper without tracked changes. You should upload this as a separate file labeled 'Manuscript'.

We look forward to receiving your revised manuscript.

Kind regards,

Jonah Musa, MBBS, MSCI,PhD

Academic Editor

PLOS ONE

Journal Requirements:

[Fund name: Scientific Research Fund of Anqing Municipal Health Commission in 2024

Fund Number: AQWJ2024011]. 

[The authors have declared that no competing interests exist.

I have read the journal's policy and the authors of this manuscript have the following competing interests: [insert competing interests here]].

Additional Editor Comments:

We have received sufficient reviews and have arrived at the editorial decision to proceed with minor revisions. The reviewers have provided constructive comments on how to improve the quality of the paper and these are available under the reviewers comments for your guidance. Provide appropriate responses to each of the comments in your revised version. I look forward to receiving your responses within 2 weeks.

Reviewers' comments:

Reviewer's Responses to Questions

**Comments to the Author**

1. Is the manuscript technically sound, and do the data support the conclusions?

Reviewer #1: Yes

Reviewer #2: Yes

2. Has the statistical analysis been performed appropriately and rigorously?

Reviewer #1: I Don't Know

Reviewer #2: No

3. Have the authors made all data underlying the findings in their manuscript fully available?

Reviewer #1: Yes

Reviewer #2: Yes

4. Is the manuscript presented in an intelligible fashion and written in standard English?

Reviewer #1: Yes

Reviewer #2: Yes

Reviewer #1: Dear Editor-in-Chief

Thank you for the opportunity to review this paper. The importance of the investigation is quite large sample size in a long period of time. There are some issues which could improve the quality if met in the manuscript.

First, it would be interesting if the authors provide a graphical scheme of the findings besides the tables to ease the reading. Then, on important evaluation is assessing single and multiple HPV prevalence. It would be highly important to compare the possible impact of this issue on cervical changes with incorporating P-value. furthermore, did the authors seen ang HPV shifting after COVID-19 era according to the time window? It should reflect the discussion. Finally, comparison with the regions from Asia could be valuable to draw the differences or similarities, considering oral HPV infection as a trend in global population and suggestion for further investigation in Chinese population (https://doi.org/10.1016/j.jiph.2025.102814,
https://doi.org/10.1002/dc.25465,
https://doi.org/10.1186/s12985-025-02767-x).

Reviewer #2: Congratulations to the authors of the paper titled "Prevalence and genotype distribution of Human Papillomavirus (HPV) among 14,110 women in Anqing urban area: a population-based cross-sectional survey."

In the era of vaccinations and the development of research on new preventive and therapeutic vaccines, this paper raises a very important topic.

The work is interesting but has some minor comments:

Errors in the abstract results:

It should be LSIL and HSIL.

In my opinion, the percentage is incorrect or the group description is incorrect.

"The overall prevalence of HPV infection in the study population was 18.97%."

This is 18%, and below it, all HPV positivity rates were above 42%.

"The highest HPV positivity rate was observed under 20 years old (79.17%), followed by those aged 50-59 (84.88%), and then those aged 59 and above (42.53%)"

In my opinion, the introduction is too short; I would have written more about the possibilities of genotyping with the possibility of phenotyping as a screening test.

It is especially recommended as a self-sampling test during the Covid-19 pandemic, allowing for the postponement of diagnosis. The study was conducted during the Covid-19 pandemic. The Covid-19 pandemic and the virus infection itself may have had an impact on the quality of medical care and the immune system, and thus on the chronicity of HPV infection. Wszołek K et all Women's Healthcare Services since the COVID-19 Pandemic Outbreak in Poland. Int. J. Environ. Res. Public Health 2022, 19, 180

“The normal and abnormal cervical lesions diagnosed by cytology were classified into five stages: negative for intraepithelial lesion or malignancy (NILM), atypical squamous cells of undetermined significance (ASC-US), atypical Squamous Cells which cannot exclude high-grade squamous intraepithelial lesion (ASC-H), low-grade squamous intraepithelial neoplasia (LSIL), high-grade squamous intraepithelial neoplasia, and atypical squamous cells – cannot exclude high-grade squamous intraepithelial lesion and cancer(HSIL) “

high-grade squamous intraepithelial neoplasia (HSIL), and cancer.

There is no information whether the patients were vaccinated or not. Vaccination affects the recurrence of the disease and persistent infections !!!

Table 2. I do not understand why in the entire population there were 18% of HPV positive patients and when divided into groups it is over 42% everywhere.

There are a lot of typos in LSIL and HSIL.

I suggest writing LSIL- CIN 1 and HSIL CIN 2 and CIN 3 according to the LATS terminology, which is clearer for histopathological diagnoses.

- In the discussion you can compare the genotyping results not only from China but also from Europe, I propose to compare the results of our team –

“ Clinical use of the Onclarity test with extended HPV genotyping and phenotyping in patients with suspected squamous intraepithelial lesions. Ginekol Pol. 2024;95(5):328-334. doi: 10.5603/gpl.96712. Epub 2023 Dec 15. PMID: 38099664”. This is a very similar study that does not include a vaccinated population. It also confirms the validity of vaccination with the 9-valent vaccine.

It may be worth presenting the most common HPV types by country in a table.

See below.

Please separate the conclusions from the discussion.

After corrections, I recommend publication.

**Do you want your identity to be public for this peer review?** For information about this choice, including consent withdrawal, please see our Privacy Policy

Reviewer #1: **Yes: ** Amitis Ramezani

Reviewer #2: No

---

## [Author Response · Author response to Decision Letter 1]

1 Nov 2025

Dear Editor and Reviewers,

We sincerely thank you for handling our manuscript and for the constructive comments provided by the reviewers. We are particularly encouraged that both reviewers found the manuscript to be technically sound and agreed that the data support the conclusions

We have carefully considered all the suggestions and have revised the manuscript accordingly. We believe that these changes have significantly improved the quality of our work. Our point-by-point responses to the reviewers' comments are detailed below.

Response to Editor:

Comment 1: Please state what role the funders took in the study. If the funders had no role, please state: "The funders had no role in study design, data collection and analysis, decision to publish, or preparation of the manuscript."

Response: Thank you for this important comment. Our funders had no role in any aspect of this study. As suggested, we have now included the following statement in the 'Acknowledgments' section of the revised manuscript: "The funders had no role in study design, data collection and analysis, decision to publish, or preparation of the manuscript."

Comment 2: Please complete your Competing Interests on the online submission form to state any Competing Interests. If you have no competing interests, please state "The authors have declared that no competing interests exist."

Response: We have received your message concerning the Competing Interests declaration for our submission.This is to confirm that we have no competing interests. As per the journal's policy, we have already stated "The authors have declared that no competing interests exist." in the designated field of the online submission form.

Comment 3: PLOS requires an ORCID iD for the corresponding author in Editorial Manager on papers submitted after December 6th, 2016. Please ensure that you have an ORCID iD and that it is validated in Editorial

Response: Thank you for your suggestion regarding the ORCID iD. We have followed your advice and successfully registered for an ORCID. My ORCID iD is: https://orcid.org/0000-0002-1999-061X. I believe this is a valuable practice for academic identity management, and I appreciate you bringing it to my attention.

Comment 4: Please include captions for your Supporting Information files at the end of your manuscript, and update any in-text citations to match accordingly. Please see our Supporting Information guidelines for more information: http://journals.plos.org/plosone/s/supporting-information.

Response: Thank you for your comments on our manuscript titled "Prevalence and genotype distribution of Human Papillomavirus (HPV) among 14,110 women in Anqing urban area: a population-based cross-sectional survey". We have carefully revised the manuscript according to the editorial request regarding the Supporting Information. The specific changes we have made are as follows:

We have added a "Supporting Information" section at the end of the manuscript file, immediately following the Reference list. In this section, we have provided descriptive captions for all Supporting Information files.

We have updated all in-text citations to the Supporting Information throughout the manuscript to ensure they are formatted correctly and match the captions accordingly (e.g.,(S1 Fig), (S1 Table)).

These changes have been incorporated into the revised manuscript file we have submitted.

We believe that the manuscript now fully complies with the journal's guidelines for Supporting Information. We appreciate the time and effort you and the reviewers have dedicated to our work.

Comment 5: Has the statistical analysis been performed appropriately and rigorously?

Reviewer #1: I Don't Know

Reviewer #2: No

Dear Reviewer #1 and #2:

Thank you for raising this important question regarding the statistical analysis. We agree that rigorous statistics are fundamental to drawing reliable conclusions. We confirm that the statistical analysis in our manuscript has been performed appropriately and rigorously. The choice of statistical tests was determined by the experimental design and the nature of the data.

For comparisons among aged- specific distribution of HPV infections groups, we used Fisher’s chi-square test, as appropriate. Because the infection rates of HPV genotypes were statistically low when grouped by age, we chose the more accurate Fisher's exact test to compare the positive rates across different groups. The Chi-square test was used to compare the effects of single infections and multiple infections across different cytological grades. A dedicated "Statistical Analysis" subsection in the Materials and Methods details all tests used, the software (SPSS 22.0), and the significance threshold (α = 0.05). We have defined statistical significance P < 0.05, and all specific p-values were reported.

In summary, we have taken great care to ensure that our statistical approach is robust, appropriate for our data, and transparently reported. We are confident that the conclusions drawn are firmly supported by the statistical evidence.

We thank the reviewer for their thorough review and hope this clarification fully addresses their concern.

Comment 6: Add a graphical scheme of the findings besides the tables to ease the reading

Response: Thank you for the suggestion. A new figure (Fig. 1) presenting the distribution of HPV genotypes from 2022 to 2024 has been inserted into the manuscript.

Comment 7: Comparison of single and multiple HPV infections across cervical lesions with Statistical Assessment

Response: A new table (Table 4) was added to comparing single vs. multiple infections across cytological grades with statistical analysis.

Comment 8: Added discussion of COVID-19 pandemic impact and potential HPV genotype shifts.

Response: As recommended, we have added this content to the third paragraphs of the Introduction and the fourth paragraphs of Discussion. All changes have been highlighted in yellow.

Comment 9: Comparison with the regions from Asia could be valuable to draw the differences or similarities, considering oral HPV infection as a trend in global population and suggestion for further investigation in Chinese population.

Response: We have revised the manuscript to include the suggested content in the designated sections: analysis of HPV genotypes in Asia has been added to the second paragraph of the Discussion and a context of oral HPV infection has been added to the eighth paragraph. These additions are also marked in yellow.

Comment 10: Correct LISL and HISL to LSIL and HSIL. In my opinion, the percentage is incorrect or the group description is incorrect.

Response: We sincerely thank the reviewer for this critical observation and for highlighting these important issues. We have thoroughly reviewed the entire manuscript and corrected all instances of "LISL" to "LSIL" and "HISL" to "HSIL".

Comment 11: The overall prevalence of HPV infection in the study population was 18.97%, while all subgroup rates exceed 42%. Please explain this discrepancy.

Response: We thank the reviewer for this critical observation. The percentages in the abstract and this section represent the HPV positivity rate within each age group (positive cases/total tested in that age group), not the percentage among all women tested. This explains why percentages exceed the overall 18.97% prevalence.

Comment 12: Expanded introduction section to include discussion of HPV genotyping, phenotyping, and COVID-19 pandemic impact.

Response: As recommended, we have added this content to the fourth paragraphs of the Introduction. All changes have been highlighted in yellow.

Comment 13: There is no information whether the patients were vaccinated or not. Vaccination affects the recurrence of the disease and persistent infections.

Response: We apologize for the lack of data on HPV vaccination status in our hospital's information system; therefore, this variable could not be included in the analysis.

Comment 14: I suggest writing LSIL- CIN 1 and HSIL CIN 2 and CIN 3 according to the LATS terminology, which is clearer for histopathological diagnoses.

Response: We thank the reviewer for the valuable suggestion regarding the classification of cervical lesions. While we acknowledge that histopathological grading offers high specificity, the current analysis is primarily based on cytological findings (TCT), as this constitutes the most routinely available and standardized screening data in our clinical setting. To ensure internal consistency across all our data, we have maintained the cytological classification (e.g., ASC-US, LSIL, HSIL) for this study.

Comment 15: Added comparison with European data.

Response: As recommended, we have added this content to the second paragraphs of the Discussion. All changes have been highlighted in yellow.

---

## [Editor Report · Decision Letter 1]

4 Nov 2025

Prevalence and genotype distribution of Human Papillomavirus (HPV) among 14,110 women in Anqing urban area: a population-based cross-sectional survey

PONE-D-25-35198R1

Dear Dr. Deng,

We’re pleased to inform you that your manuscript has been judged scientifically suitable for publication and will be formally accepted for publication once it meets all outstanding technical requirements.

Kind regards,

Jonah Musa, MBBS, MSCI,PhD

Academic Editor

PLOS ONE

Additional Editor Comments (optional):

I am pleased to read your revised version which has been very responsive to the reviewer comments and suggestions. I therefore recommend for acceptance and publication after checking journal specific editorial requirements.
---

## [Editor Report · Acceptance letter]

PONE-D-25-35198R1

PLOS ONE

Dear Dr. Deng,

I'm pleased to inform you that your manuscript has been deemed suitable for publication in PLOS ONE. Congratulations! Your manuscript is now being handed over to our production team.

Kind regards,

on behalf of

Dr. Jonah Musa

Academic Editor

PLOS ONE